# Dynamic of Mayaro Virus Transmission in *Aedes aegypti*, *Culex quinquefasciatus* Mosquitoes, and a Mice Model

**DOI:** 10.3390/v15030799

**Published:** 2023-03-21

**Authors:** Larissa Krokovsky, Carlos Ralph Batista Lins, Duschinka Ribeiro Duarte Guedes, Gabriel da Luz Wallau, Constância Flávia Junqueira Ayres, Marcelo Henrique Santos Paiva

**Affiliations:** 1Departamento de Entomologia, Instituto Aggeu Magalhães, Fundação Oswaldo Cruz, Av. Professor Moraes Rego, S/N, Campus da UFPE, Cidade Universitária, Recife 50740-465, PE, Brazil; 2Biotério de Criação, Instituto Aggeu Magalhães, Fundação Oswaldo Cruz, Av. Professor Moraes Rego, S/N, Campus da UFPE, Cidade Universitária, Recife 50740-465, PE, Brazil; 3Núcleo de Ciências da Vida, Centro Acadêmico do Agreste, Universidade Federal de Pernambuco (UFPE), Rodovia BR-104, km 59-Nova Caruaru, Caruaru 55002-970, PE, Brazil

**Keywords:** alphavirus, Culicidae, virus–vector interaction, vector competence studies, IFNAR BL/6 mice

## Abstract

Mayaro virus (MAYV) is transmitted by *Haemagogus* spp. mosquitoes and has been circulating in Amazon areas in the North and Central West regions of Brazil since the 1980s, with an increase in human case notifications in the last 10 years. MAYV introduction in urban areas is a public health concern as infections can cause severe symptoms similar to other alphaviruses. Studies with *Aedes aegypti* have demonstrated the potential vector competence of the species and the detection of MAYV in urban populations of mosquitoes. Considering the two most abundant urban mosquito species in Brazil, we investigated the dynamics of MAYV transmission by *Ae. aegypti* and *Culex quinquefasciatus* in a mice model. Mosquito colonies were artificially fed with blood containing MAYV and infection (IR) and dissemination rates (DR) were evaluated. On the 7th day post-infection (dpi), IFNAR BL/6 mice were made available as a blood source to both mosquito species. After the appearance of clinical signs of infection, a second blood feeding was performed with a new group of non-infected mosquitoes. RT-qPCR and plaque assays were carried out with animal and mosquito tissues to determine IR and DR. For *Ae. aegypti*, we found an IR of 97.5–100% and a DR reached 100% in both 7 and 14 dpi. While IR and DR for *Cx. quinquefasciatus* was 13.1–14.81% and 60% to 80%, respectively. A total of 18 mice were used (test = 12 and control = 6) for *Ae. aegypti* and 12 (test = 8 and control = 4) for *Cx. quinquefasciatus* to evaluate the mosquito–mice transmission rate. All mice that were bitten by infected *Ae. aegypti* showed clinical signs of infection while all mice exposed to infected *Cx. quinquefasciatus* mosquitoes remained healthy. Viremia in the mice from *Ae. aegypti* group ranged from 2.5 × 10^8^ to 5 × 10^9^ PFU/mL. *Ae. aegypti* from the second blood feeding showed a 50% IR. Our study showed the applicability of an efficient model to complete arbovirus transmission cycle studies and suggests that the *Ae. aegypti* population evaluated is a competent vector for MAYV, while highlighting the vectorial capacity of *Ae. aegypti* and the possible introduction into urban areas. The mice model employed here is an important tool for arthropod–vector transmission studies with laboratory and field mosquito populations, as well as with other arboviruses.

## 1. Introduction

Mayaro virus (MAYV) is an arbovirus from the *Togaviridae* family and the *Alphavirus* genus that was first identified in Trinidad in 1954 [1]. Since then, MAYV has been found in several countries in Latin America, mostly in regions of the Amazon area [2,3,4,5]. In Brazil, the virus has been circulating for decades and has recently caused outbreaks in the Northern and Central West regions of the country [6]. Symptoms caused by MAYV are not specific and can range from fever, chills, gastrointestinal symptoms, retro orbital pain, myalgia, and arthralgia [7,8]. As with the Chikungunya virus, MAYV causes arthralgia in 50 to 90% of patients and myalgia affects 75% of infected humans [9]. In particular, arthralgia can last from months to years and severe complications such as myocarditis, and hemorrhagic and neurological manifestations can also affect patients [10]. MAYV surveillance in Brazil was recently intensified, mainly due to the need for differential diagnosis for alphaviruses presenting similar clinical symptoms and the limited diagnostic laboratory facilities in many of the endemic regions [11].

MAYV has a sylvatic transmission cycle maintained between *Haemagoggus* spp. and non-human primates as vertebrate hosts. However, other species, from genera *Mansonia*, *Culex*, *Sabethes*, and *Aedes* and vertebrates such as rodents and reptiles may be involved in the transmission cycle [12,13,14,15]. A concern is the increase in vector-competent mosquito species feeding on susceptible vertebrates found in urban areas ultimately introducing the virus into cities. Vector competence for MAYV has already been demonstrated in *Aedes aegypti* and *Aedes albopictus* [16,17,18], while *Culex quinquefasciatus* species is an inefficient vector for this virus [17]. Studies with field-caught mosquitoes showed eight genera naturally infected with MAYV including mosquitoes from *Ae. aegypti*, *Ae. albopictus,* and *Cx. quinquefasciatus* species [12,19,20]. Vertical transmission in *Ae. aegypti* was demonstrated with the detection of MAYV in eggs collected in the field [21]. The transmission cycle dynamics of MAYV involving different species of mosquitoes and vertebrate hosts are still unclear. Therefore, vector competence studies addressing the vector–host cycle are needed to broadly assess different mosquito populations and viral strains since a set of environmental and genetic factors of viruses and vectors can alter the potential viral spread. Currently, there are no published data focused on infection and transmission of MAYV from vectors to animals [10].

In studies on pathogenesis and vaccine, experimental animal models used for MAYV are mainly murine models in three categories: the neonatal model of lethal challenge, immunosuppressed models of lethal disease, and the arthritis model [22,23,24]. The increased circulation of MAYV makes it necessary to establish a new study model with an experimental focus on the transmission cycle and pathogenicity related to the chronic symptoms of the infection [10]. Artificially feeding mosquitoes with MAYV or mice infection is a good alternative to evaluate parameters such as vector competence and pathogenicity in mammals. However, experimental methods do not necessarily represent the complexity of a blood meal in a live host, which potentially contains host immunological factors and viral antigens that can influence mosquito infection and virus transmission [25,26,27,28]. In vector competence studies, there are also gaps and a lack of standardization of the methods and nomenclatures used in the assays [29]. Investigation with an animal model and stages of the transmission cycle should be performed with the aim of improving our understanding of pathogen–vector–host interaction dynamics. 

Considering the two most abundant species in the urban environments in Brazil, the aim of the present study was to evaluate the transmission dynamics of MAYV by *Ae. aegypti* and *Cx. quinquefasciatus* from Recife-PE using an animal model (mosquito–host–mosquito). The focus of the study was to characterize the virus–vector–host interaction for MAYV, since a complete transmission model for MAYV has not yet been described, and the imminent risk of emergence of the virus through species such as *Ae. aegypti*.

## 2. Materials and Methods

### 2.1. Cells and Viral Strain

VERO CCL81 cells were grown in Minimum Essential medium (MEM; Gibco, Grand Island, NY, USA) supplemented with 10% fetal bovine serum (FBS; Gibco, Grand Island, NY, USA) and 1% penicillin/streptomycin (Gibco, Grand Island, NY, USA) in a 37 °C + 5% CO_2_ incubator_._ The MAYV strain used for mosquito infection was MAYV/BR/Sinop/H307/2015 (MH513597) derived from a human serum sample kindly provided by Dr. Roberta Bronzoni from the Federal University of Mato Grosso (UFMT-Sinop). Viral stocks were prepared by inoculation on to an 80–90% confluent monolayer of VERO cells. Virus culture were harvested at 24 h after infection and centrifuged at 1200 rpm for 10 min and supernatants were transferred to a new cryotube and then stored at −80 °C until use. Prior to mosquito infection, viral titer was calculated via plaque assay and reached 10^8^ plaque-forming units per milliliter (PFU/mL).

### 2.2. Mosquitoes and Animals

Mosquito colonies used in the present study were derived from two natural populations collected in Recife/PE, Brazil: RecLab (*Ae. aegypti*) and CqSLab (*Cx. quinquefasciatus*). These mosquito colonies have been maintained at standard conditions of 26 ± 2 °C, 65–85% relative humidity, and 10/14 light/dark cycle in the Entomology Department at the Aggeu Magalhães Institute (AMI) for several generations [30,31].

Concerning animal models for virus transmission, we used an IFNAR BL/6 (-Ifnar1^tm1.2Ees^/J) mice strain which was provided by the Jackson Laboratory Repository (JAX, Bar Harbor, ME, USA) to the Central Animal Facility of the University of São Paulo, School of Medicine, in Ribeirão Preto/SP, Brazil. This center kindly provided the animals to the AMI Animal Facility installation with the required sanitary and genetic certification.

### 2.3. Mosquitoes Artificial Feeding and Vector Competence Evaluation 

Seven-to-ten-day-old mosquitoes from RecLab and CqSLab colonies were separated into different plastic cages named test and control groups for each species. The test group contained 200 females and 50 males, while the control group contained 100 females and 30 males. Twenty-four hours prior to artificial blood feeding, both species were sugar starved. Artificial blood-feeding assay was carried out in a BSL2 (Biosafety Level 2) and then all the plastics cages were maintained in containment cages. The experiments with the test group were conducted using MAYV obtained from VERO cells collected 24 h post-inoculation at a multiplicity of infection of 0.1, while uninfected cultured cells were used for the control group. The cell culture flasks were frozen at −80 °C, thawed at 37 °C, and then mixed with defibrinated rabbit blood in a 1:1 proportion and ATP (Adenosine Triphosphate) (Sigma-Aldrich, San Luis, MO, USA) in a 3 mM final concentration. The blood mixture was then offered in 4 cm diameter Petri dishes containing a triple membrane of PARAFILM^®^ M (Sigma-Aldrich, San Luis, MO, USA) on top of the cages at 37 °C by using heat packs for 1 hour. Fully engorged females were cold anesthetized, transferred to a new cage, and maintained in the infection room for 14 days post-infection (dpi). Blood mixtures used in the assays as well as fully engorged females were collected and evaluated as artificial control samples.

In order to investigate vector competence of *Ae. aegypti* and *Cx. quinquefasciatus* previously fed with MAYV, 15 females for each species were collected at 3 dpi, 7 dpi, and 14 dpi. Vector competence was estimated by the detection of MAYV in dissected midguts (MID) and salivary glands (SG) at different time points (Figure 1A). Tissues were collected in a Petri dish containing cold PBS 1X (Thermo Fisher Scientific, Waltham, MA, USA) with a forceps and visualization was performed in a stereomicroscope (LABOMED, Los Angeles, CA, USA). Dissection was performed quickly on ice, and before every mosquito dissection, all instruments were cleaned using a 70% ethanol. Immediately after dissection, tissues were individually transferred to a 1.5 mL DNase/RNAse-free microtube containing 300 μL of mosquito diluent (1X PBS, 20% FBS, 1% penicillin/streptomycin) [32] and stored at −80 °C until further use.

### 2.4. MAYV Transmission Cycle Model in Mice

In order to determine whether the viral load detected in mosquitoes is sufficient to infect naïve mice and then infect new susceptible mosquitoes (complete transmission cycle), transmission dynamics assays were performed with IFNAR BL/6 mice aged between 2–4 weeks, males and females, separated in micro-isolators into test groups comprising 4 animals and control groups comprising 2 animals. The animals were kept at 22 ± 2 °C, 50–60% relative humidity, 12/12 light/dark cycle, and were acclimated for at least 72 h before use. The time point (3 dpi, 7dpi, or 14 dpi) for the analysis of animal transmission by the mosquitoes infected with MAYV was evaluated after vector competence assays. At the chosen time point when mosquito colonies showed high dissemination rates, both animal groups (test and control) were weighed, anesthetized with the association of Ketamine (100 mg/kg) (Syntec, Barueri, São Paulo, Brazil) and Xylazine (10 mg/kg) (Syntec, Barueri, São Paulo, Brazil), and then placed on the surface of plastic cages containing mosquitoes (10–15 females per mice) for 20 min. During the procedure, the animals were monitored for any signs of suffering. After the blood meal, the animals were monitored daily for seven days to assess weight loss, clinical signs, and mortality rate (Figure 1B). Numerical values were assigned to the clinical signs according to the following score: isolation—1, ruffled hair—1, mild lethargy—1, curved posture—2, moderate lethargy—2, weight loss (≥20%)—2, foot edema—2, severe lethargy—3, and diarrhea—3. 

At the moment that signs of infection were identified (viremia period considered with score ≥ 3), the animals from the test group were again anesthetized to be available as a blood source for a new group of naïve mosquitoes (10–15 females per mice) according to the protocol described above. After the blood meal in infected mice, mosquitoes were kept in the infection room for another 7 days and then collected, anesthetized at −20 °C, and individually stored in 1.5 mL RNAse/DNAse-free tubes containing 300 μL of diluent [32] (Figure 1B).

After the second blood meal with naïve mosquitoes was performed, whole blood was collected from each mouse and euthanasia was carried out by cardiac puncture. After the procedure, mice were dissected to collected brain, liver, and gonads from control and test groups using sterile surgical scissors and forceps. Tissues were stored in 2 mL RNAse/DNAse-free tubes containing 500 μL of diluent [32]. Blood was centrifuged for 10 min at 4000 rpm and serum was transferred to new 1.5 mL RNAse/DNAse-free tubes.

### 2.5. RNA Isolation and RT-qPCR

Mosquito and mice tissues were individually homogenized as described by Barbosa et al. [33]. RNA extraction was performed with TRIzol^®^ reagent (Invitrogen, Carlsbad, CA, USA) as previously described [34]. Each RT-qPCR reaction was carried out using the QuantiNova Kit Probe RT-PCR Kit (Qiagen, Hilden, North Rhine-Westphalia, Germany) using primers and a probe, as described elsewhere [35]. The reactions were performed in a final volume of 10 μL with the following final concentrations: QuantiNova Probe RT-PCR Master Mix (1×), QuantiNova ROX (1×), QuantiNova RT Mix (1×), 800 nM of primers and 100 nM of probes. RT-qPCR reactions were performed using a QuantStudio 5 System (Applied BioSystems, Norwalk, CT, USA) under the following conditions: 45 °C for 15 min, 95 °C for 5 min, followed by 45 cycles of 95 °C for 5 s and 60 °C for 45 s. All samples were tested in duplicates, with negative reaction control (all reagents except for RNA), negative control from RNA isolation and positive control (standard curve). The standard curve used was synthesized by in vitro transcription using MEGAscript T7 kit (Ambion, Austin, TX, USA) and was quantified in Nanodrop 2000. RNA concentration was converted in an RNA copy number as previously described [36]. These results were analyzed using QuantStudio Design and Analysis Software v. 1.3.1 with automatic threshold and baseline. Samples with cycle quantification (Cq) values ≤ 38.5 in both duplicates were considered as positive.

### 2.6. Plaque Assay

Viral titer in mosquito and mice tissues was calculated using plaque assay. The 24-well plates containing 3 × 10^5^ cells/mL in MEM supplemented with 10% FBS were prepared and incubated at 37 °C + 5% CO_2_. After 24 h, the medium was discarded and 50 μL of a serial dilution (10^−1^ to 10^−10^) of each sample in duplicate was added to the monolayer and incubated for 30 min at 37 °C + 5% CO_2_ for adsorption. After incubation, 500 μL of MEM semi-solid (3% carboxymethylcellulose, Sigma-Aldrich, San Luis, MO, USA) was added. After 48 h of incubation at 37 °C + 5% CO_2_, the medium was removed by inversion, and the cell monolayer was fixed with 8% formalin solution (Sigma-Aldrich, St. Louis, MO, USA) for 1 hour and 20 min and then revealed with crystal violet 0.04% (Sigma-Aldrich, San Luis, MO, USA) for plaque visualization and counting.

### 2.7. Data Analysis

The infection rate (IR = positive midguts/midguts analyzed) and dissemination rate (DR = positive salivary glands/positive midguts) were calculated for each species at different time points. Logistic regression, Χ^2^-test, and Fisher’s Exact tests were used to test differences in both the IR and DR for the two species. The Wilcoxon test was used to compare MID and SG values. One-way analysis of variance (ANOVA) followed by Tukey’s multiple comparisons test was used for mice tissues analysis. Analyzed results were considered as significant when the *p* value < 0.05. All statistical tests and graphics were performed with R software (R DEVELOPMENT CORE TEAM) and Graph Pad Prism 9 software (Graph Pad, San Diego, CA, USA).

### 2.8. Ethic Statement

This project was approved by the Ethics Committee for Animal Use of the AMI, under the protocol CEUA/IAM 166/2021, and followed the guidelines of the National Council for the Control of Animal Experimentation (CONCEA).

## 3. Results

### 3.1. Vector Competence Assays

To evaluate vector competence of *Ae. aegypti* (RecLab) and *Cx. quinquefasciatus* (CqSLab) colonies for MAYV, three independent artificial blood feeding experiments were performed for each species. The titer of the initial mixture (MAYV culture + blood) before artificial blood feeding ranged from 2.5 × 10^6^ to 1.5 × 10^7^ PFU/mL. MID and SG of 135 females of *Ae. aegypti* and 100 females of *Cx. quinquefasciatus* were analyzed (Table 1). On the 3rd, 7th, and 14th dpi the tissues were dissected and tested by RT-qPCR to verify the presence and to quantify MAYV. For *Ae. aegypti,* the IR found were 97.7% (3 dpi), 100% (7 dpi), and 100% (14 dpi), and the DR reached 100% in both 7 and 14 dpi. Regarding *Cx. quinquefasciatus*, the IR found on the 3rd, 7th, and 14th dpi were 14.28%, 13.1%, and 14.81%, respectively. In the CqSLab colony, even with a low IR, 10 positive salivary glands were detected, which generated a DR that ranged between 60% and 80% at the time points (Figure 2).

Cq values from RecLab colony showed significant lower Cq values in MID (ranged from 17 to 31.7) when compared with CqSLab (ranging from 29 to 37.5) with *p* < 0.0001 at 3dpi, *p* < 0.001 at 7 dpi, and *p* < 0.0001 at 14 dpi. RecLab also showed lower SG Cq values (ranging from 17 to 37.6) when compared with CqSLab (ranging from 25.5 to 38) with *p* < 0.05 for all time points. In terms of RNA copy number/mL, the quantification showed the same pattern on the MID samples of both species; however, the SG quantification showed a difference only at the 7 dpi (*p* < 0.0001) (Figure 3A,B). Plaque assays were performed with 12 GS samples from RecLab (Cq range from 17.2 to 37.5) and samples with Cq < 30 showed plaque-forming units that ranged from 2.5 × 10^3^ to 1.25 × 10^6^ PFU/mL. All CqSLab MID and SG samples (*N* = 24) were submitted to a plaque assay and only one SG sample (Cq = 25.2) formed plaque and showed a titer of 1.25 × 10^4^ PFU/mL.

### 3.2. Mayaro Virus Transmission Cycle in Mice

After analysis of infection and dissemination rates in both mosquito species, we followed up MAYV transmission rate to mice on day 7 post-infection using the RecLab colony as a reference, as this species showed the optimal results for virus replication in salivary glands. Cq values and RNA copy number/mL in *Ae. aegypti* showed higher amounts of virus at 7 dpi (*p* < 0.001).

Three independent experiments were performed for MAYV transmission in a mouse model. A total of 30 mice were used in the experiments: 12 animals from the test group and six animals from the control group with RecLab mosquitoes, and eight animals of the test group and four animals of the control group with CqSLab mosquitoes (Table 2). A total of eight mice infected by *Ae. aegypti* demonstrated signs of infection at 3 dpi and, in addition, four animals died at 4 dpi. Viral load was calculated in infected mice serum and ranged from 2.5 × 10^8^ to 5 × 10^9^ PFU/mL (Table 2). Animals presented with a classic sign of alphaviruses, foot edema, as well as several other signs of infection (Figure 4A,B). The scores observed in mice are described in Table 2. No mice from the infection group nor any from the control group showed sign of infection in the CqSLab experiments.

Brain, liver, and gonad tissues of all animals were collected, including those that succumbed to the infection and died. In summary, all organs that were processed by RT-qPCR of the control groups were negative; in the CqSLab test group, out of 24 tissue samples collected, only one brain was positive with a Cq value of 32.9. Animals from the RecLab test group showed MAYV presence in all tissues. Organs of the infected animals were also assayed by a plaque assay, and all samples presented plaque-forming units (Figure 4C). Cq values ranged 17.5 to 24.3 (brain), 12.3 to 18.5 (liver), and 10.7 to 21.8 (gonads). Regarding viral load (PFU/mL), titers ranged from 2.5 × 10^5^ to 1 × 10^7^ in the brain, 5 × 10^6^ to 3.5 × 10^8^ in the liver, and 1.5 × 10^5^ to 2.5 × 10^7^ on gonads. Statistical analysis was performed on Cq values, RNA copy number, and viral load (PFU/mL). For Cq values, the results showed that the brain was the organ with the highest Cq value when compared with the other two organs. When we looked at the quantification of RNA copy number/mL, we found that the liver was the tissue with the highest RNA copy number/mL when compared with the brain and gonads (Figure 4E). The MAYV tropism for the liver was confirmed as we evaluated the virus load (PFU/mL) found in this particular tissue (Figure 4D).

After visualization of signs of infection at third day, animals from the test group exposed to *Ae. aegypti* (RecLab) were available as a blood source for mosquitoes from the same species. For the CqSLab group, we waited until the seventh day and then used mice as the blood source for another group of naïve mosquitoes from the same species. At day 7 after blood feeding on mice, all mosquitoes (RecLab and CqSLab) were collected and 30 individuals (whole body) were assayed by RT-qPCR. As a result, concerning *Cx. quinquefasciatus,* no mosquito was found to be positive. *Ae. aegypti* samples showed 15 out of 30 (50%) mosquitoes positive for MAYV and Cq values ranged from 17.6 to 24.8 and the quantification of the RNA copy number/mL is described in Figure 4E.

## 4. Discussion

Arbovirus are maintained in nature through sylvatic and urban transmission cycles involving hematophagous invertebrate vectors and a range of vertebrate hosts. These viruses are responsible for outbreaks and epidemics all over the world. MAYV is considered as an emerging arbovirus and an increasing number of human cases has been reported in several Latin American countries [10,12]. Despite the importance, there is an absence of pathogen–vector–host interaction studies for MAYV, highlighting the urgent need for such studies, especially in developing countries, which lack resources for arbovirus transmission control. Here, we demonstrated a complete model of MAYV transmission (mosquito–host–mosquito) with two mosquito species, which are very abundant in urban areas (*Ae. aegypti* and *Cx. quinquefasciatus*). We showed high IR and DR for *Ae. aegypti* and low IR and high DR for *Cx. quinquefasciatus*. After analysis of the vector competence of both species, mosquitoes were used to test the transmission of MAYV to naïve mice. *Ae. aegypti* were able to transmit and infect 100% of the mice available for feeding, and naïve mosquitoes of this species were also able to become infected and maintain a 50% infection rate after seven days post-blood feeding on MAYV-infected mice.

Regarding the vector competence data, we found a high IR (>97%) and DR (>61%) for *Ae. aegypti*, in line with findings by Kantor et al. [37], using *Ae. aegypti* populations from the USA (>80%), and Long et al. [38], who used *Ae. aegypti* from Peru (>80%). Pereira et al. [17] performed experimental infection assays with field populations of *Ae. aegypti*, *Ae. albopictus,* and *Cx. quinquefasciatus* from Minas Gerais and detected infection rates of 57.5%, 61.6%, and 2.5% respectively. For *Ae. aegypti* and *Ae. albopictus*, results are similar to those observed in the present study; however, IR was much lower in *Cx. quinquefasciatus* when compared with our results of 13–15%. While the DR for *Cx. quinquefasciatus* found in the cited study was zero, our data showed 10 positive salivary glands for MAYV RNA, which generated a DR of 60 to 80%. However, when *Cx. quinquefasciatus* samples were inoculated into VERO cell monolayers, only one sample formed plaques, likely indicating a low proportion of infectious particles in the sample that resulted in limited active viral replication.

With the animal model proposed here, we demonstrated the ability of *Ae. aegypti* to transmit MAYV to 100% of susceptible mice and then the ability of those viremic animals to transmit the virus to mosquitoes after three days. Some studies used saliva collection as a parameter of virus transmission [17,39]; however, the detection of viruses in saliva is not an effective method. In this approach, there is a methodological difficulty in confirmation of saliva collection and processing a small amount of saliva samples. Some studies addressed vector competence with different methodological approaches, Secundino et al. [40] evaluated the transmission of ZIKV by *Ae. aegypti,* allowing the mosquitoes to feed on the mice’s ear. Long et al. [38] used vertebrate hosts to assess the transmission of MAYV by *Ae. aegypti*. Hibl et al. [41] developed a study with a new model of infection that included the natural vectorial transmission of CHIKV by *Ae. aegypti* to a humanized mouse model and Weger-Lucarelli et al. [42] also used an animal model with transmission by mosquitoe bites and focused on diet and obesity during MAYV infection. 

Multiple animal models have been used to study the pathogenesis and tropism of Alphavirus, such as CHIKV and MAYV [43,44]. We demonstrated classic signs of infection such as footpad edema and a short incubation period (3–4 days after infection) in the animal model used in this study, which corroborates the findings of Santos et al. [45] and Marín-Lopez et al. [43]. Acute infection by alphaviruses and high mortality rate of immunocompromised mice were also demonstrated by Seymour et al. [46] in O’nyong-nyong virus infection, and they were detected in the current study. Regarding virus tropism to different tissues, our data showed higher virus titer in the liver when compared with the brain and gonads; these results corroborate the findings described by Marín-Lopez et al. [43]. In contrast to what has been demonstrated with MAYV in our study, the opposite is seen for infection by ZIKV, where replication increased in organs such as the brain and caused severe disease in the central nervous system [47,48].

The understanding of the transmission process by mosquitoes and infection of the vertebrate host have relevance for public health, especially in the case of arboviruses being introduced into urban environments and causing outbreaks. In the present study, we described a model that can be used to evaluate vector competence of different arboviruses, pathogenesis in mammals infected similarly to natural viral transmission, and also for future studies of viral genetic evolution. Grubaugh et al. [49] used chickens to evaluate the replication, diversity, and evolution of the West Nile virus transmitted by *Culex* mosquitoes and showed a diversification and evolution of the virus in the different stages of the transmission cycle. Studies in this direction can help to elucidate the vector competence of some mosquito species and the adaptation of some viruses in specific environments [50].

Vector competence studies have been published for decades; however, these studies have shown insufficient data that are lacking detail or discrepant conclusions. In studies with vectors and vector–pathogen interactions, it is important to focus on the standardization of methods and nomenclature [29]. The study design described here has the potential to be widely used with mosquitoes and arboviruses and reinforces the importance of conducting vector competence studies with a complete transmission cycle. An animal model also expands the possibility of evaluating more generalist mosquito species in blood preference that are not easily fed artificially. In addition to studies of infection by one virus, our model can also be used in co-transmission and coinfection assays. A detailed understanding of how these scenarios impact virus transmission, as well as pathogenicity, is essential owing to the current simultaneous co-circulation of several arboviruses in Brazil and other endemic countries.

## 5. Conclusions

We demonstrated here that when *Ae. aegypti* mosquitoes were infected with MAYV, they were capable of transmitting the virus to susceptible mice that develop signs of the disease. These same mice were also able to serve as an infectious blood meal to another group of naïve *Ae. aegypti*, completing the virus transmission cycle: vector–host–vector. On the other hand, mice exposed to MAYV-infected *Cx. quinquefasciatus* remained healthy. The dynamics of MAYV infection showed here can be used with different arboviruses and mosquito species for a better understanding of the transmission cycle. 

## Figures and Tables

**Figure 1 viruses-15-00799-f001:**
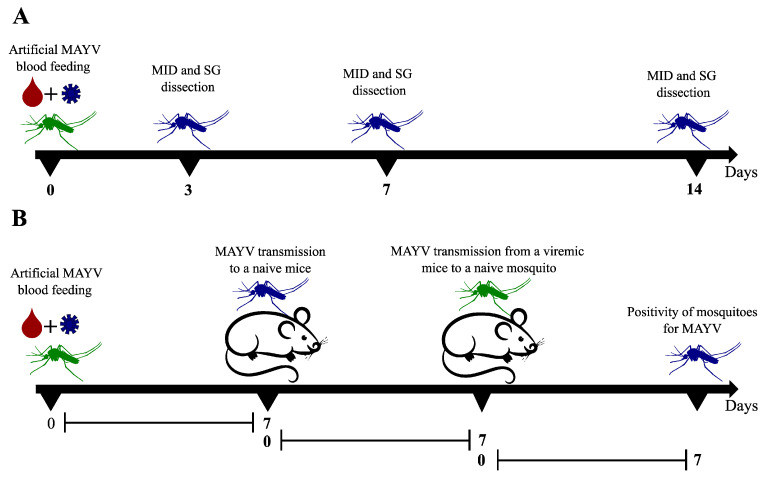
Illustration of the vector competence study design of Mayaro virus transmission in a mouse model. (**A**) Vector competence study design in a timeline. (**B**) Mayaro transmission cycle model in a timeline. MID—midguts; SG—salivary glands; and MAYV—Mayaro virus. The illustration was created by the authors using the software Inkscape v 1.2.

**Figure 2 viruses-15-00799-f002:**
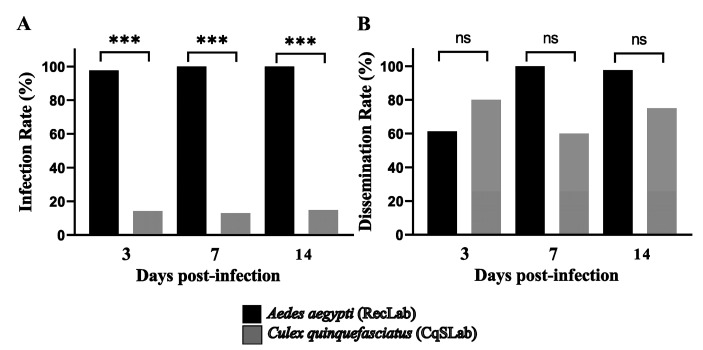
Infection and dissemination rates of *Aedes aegypti* (RecLab) and *Culex quinquefasciatus* (CqSLab) mosquitoes experimentally fed with blood containing Mayaro. (**A**) Representative graph of infection rates found in *Aedes aegypti* (RecLab) and *Culex quinquefasciatus* (CqSLab). (**B**) Representative graph of dissemination rates found in *Aedes aegypti* (RecLab) and *Culex quinquefasciatus* (CqSLab). Dpi—day post-infection; ns—not significant. Statistical analysis was performed using R software (R DEVELOPMENT CORE TEAM) (*** *p* < 0.001).

**Figure 3 viruses-15-00799-f003:**
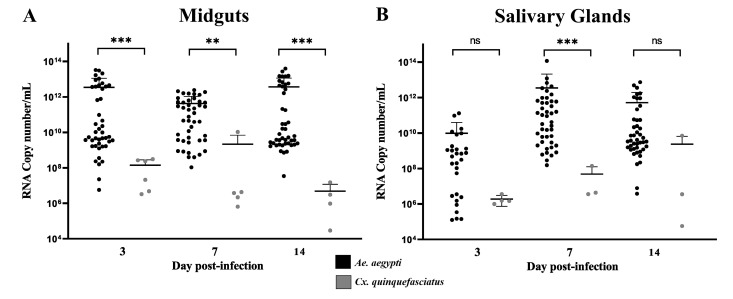
Positive midgut and salivary glands of *Aedes aegypti* and *Culex quinquefasciatus* by RT-qPCR for Mayaro virus. (**A**,**B**) Quantification of RNA viral copy numbers in the positive midguts and salivary glands of *Aedes aegypti* and *Culex quinquefasciatus* mosquitoes experimentally fed with blood containing MAYV. Dpi—day post-infection; ns—not significant. Statistical analysis was performed using the R software (R DEVELOPMENT CORE TEAM) ** *p* < 0.01, and *** *p* < 0.001).

**Figure 4 viruses-15-00799-f004:**
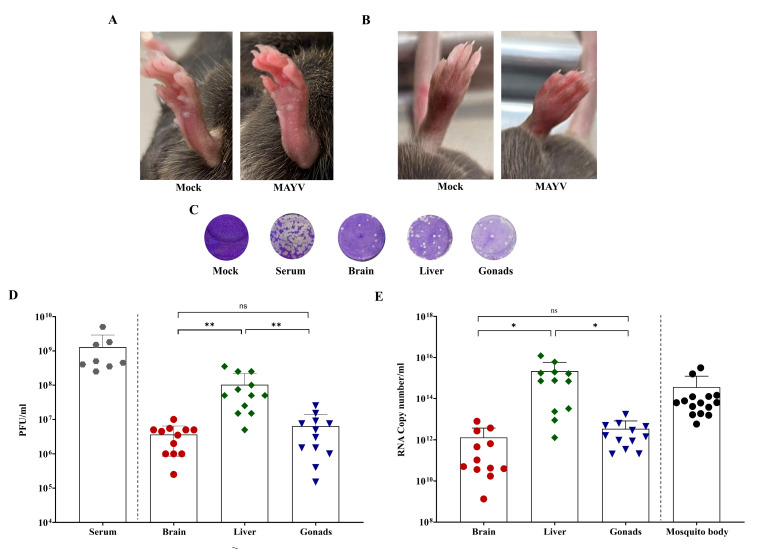
Animal experimentation panel with results of IFNAR BL/6 mice infected with Mayaro and mosquitoes *Aedes aegypti*. (**A**) Foot edema presented in mice hind foot after Mayaro virus infection compared with mock hind foot showed in bottom view; (**B**) foot edema presented in mice hind foot after Mayaro infection and mock hind foot showed in top view; (**C**) visualization of plaque-forming units in different mice samples at 10^−5^ dilution in VERO cell plaque assay; (**D**) Mayaro titer in mice tissues infected with Mayaro virus after *Aedes aegypti* blood feeding; (**E**) RNA copy number of mice tissues and *Aedes aegypti* whole-body sample infected with Mayaro virus after *Aedes aegypti* blood feeding. Gray circles—mice serum samples; Red circles—mice brain samples; Green diamond—mice liver samples; Blue triangle—mice gonad samples; Black circles—mosquito body samples; ns—not significant. Statistical analysis was performed using Graph Pad Prism 9 (* *p* < 0.05, ** *p* < 0.01).

**Table 1 viruses-15-00799-t001:** Description of positive samples, infection rate, and dissemination rate for Mayaro virus from *Aedes aegypti* (RecLab) and *Culex quinquefasciatus* (CqSLab) after artificial blood feeding with Mayaro virus.

Species	Dpi	MID +	IR	Cq Range	SG +	DR	Cq Range
*Ae. aegypti* RecLab	3	44/45	97.7%	17.8–31.5	27/44	61.3%	20.2–37.6
	7	45/45	100%	17–25.5	45/45	100%	17–32.2
	14	45/45	100%	17.5–31.7	44/45	97.7%	20–35.1
		134/135			116/134		
*Cx. quinquefasciatus* CqSLab	3	5/35	14.28%	29–36.8	4/5	80%	35.8–37.8
	7	5/38	13.1%	31–36.7	3/5	60%	35.2–38
	14	4/27	14.81	35–37.5	3/4	75%	25.2–37.3
		14/100			10/14		

Dpi—day post-infection; MID—midguts; IR—infection rate; Cq—cycle quantification; SG—salivary gland; and DR—dissemination rate.

**Table 2 viruses-15-00799-t002:** Description of IFNAR BL/6 mice used in Mayaro transmission cycle experiments with *Aedes aegypti* and *Culex quinquefasciatus* mosquitoes.

Species	Group	Mice	Sex	Assay	Clinical Score	Dpi	Viremia (PFU/mL)
*Ae. aegypti* RecLab	Control	1	M	1	0	3	N.A.
		2	M	1	0	3	N.A.
		3	F	2	0	4	N.A.
		4	F	2	0	4	N.A.
		5	F	3	0	3	N.A.
		6	F	3	0	3	N.A.
	Test	1	M	1	5	3	4 × 10^8^
		2	M	1	7	3	1.5 × 10^9^
		3	M	1	5	3	5 × 10^9^
		4	M	1	5	3	1.8 × 10^9^
		5	F	2	Mortality	4	N.A.
		6	F	2	Mortality	4	N.A.
		7	F	2	Mortality	4	N.A.
		8	F	2	Mortality	4	N.A.
		9	F	3	7	3	4.5 × 10^8^
		10	F	3	7	3	2.5 × 10^8^
		11	F	3	5	3	5 × 10^8^
		12	F	3	5	3	3.5 × 10^8^
*Cx. quinquefasciatus* CqSLab	Control	1	F	1	0	7	N.A.
		2	F	1	0	7	N.A.
		3	M	2	0	7	N.A.
		4	M	2	0	7	N.A.
	Test	1	F	1	0	7	N.A.
		2	F	1	0	7	N.A.
		3	F	1	0	7	N.A.
		4	F	1	0	7	N.A.
		5	M	2	0	7	N.A.
		6	M	2	0	7	N.A.
		7	M	2	0	7	N.A.
		8	M	2	0	7	N.A.

Dpi—day post-infection; M—male; F—female; N.A.—not applicable.

## Data Availability

Data sharing not applicable.

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
