# Peer review of "Dynamic of Mayaro Virus Transmission in *Aedes aegypti*, *Culex quinquefasciatus* Mosquitoes, and a Mice Model"

_viruses, 2023, doi:10.3390/v15030799_

Round 1

Reviewer 1 Report

In this study the authors reported on the transmission cycle of MAYV, and alphavirus endemic to Brazil. The authors made use of laboratory reared mosquitoes to demonstrate the oral susceptibility and vector status for MAYV for Ae. Aegypti but not for Cx quinquefasciatus mosquitoes. They then demonstrated transmission of the virus to mice that was able to produce replicating virus, as shown with PFU assay, and transmit the virus back to naïve mosquitoes. 

The methods and techniques displayed in this manuscript is applauded. Scientifically this work would greatly contribute to vector competence studies especially for more general vectors which are usually overseen. The use of multiple assays to confirm results contributes to the strength of this paper.

What was interesting was the presence of virus in the brain since MAYV is an old-world virus. Maybe the authors could add a sentence in the discussion for possible reasons for this with some literature.

Although the science is sound, the language use is poor and extensive English editing is needed. I have made some suggestions and highlighted areas that can be deleted. Please see the pdf document.

More specific:

Overall, the abstract can be improved

Figure 4 is not necessary.

Figure 5: It would be interesting to see the actual replication/titer compared to the initial virus that was used for the feeding (mosquito at day 7).

Please explain line 382-387 (and figure 5)

Did the mosquitoes feed on the mice 3 and 7 dpi with MAYV and then after 7 days the mosquitoes were tested? So figure 5G is a combination of days 3 and 7 feeding? Why merge the days? If there’s a difference/no difference its important with regards to incubation time in mice.

Overuse of regarding and concerning at the start of sentences

In the discussion, please be more clear when mentioning previous work done. 

Author Response

Author's comment: All suggestions found in the manuscript file were accepted and incorporated to the text.

Point by point answers:

Overall, the abstract can be improved

Answer:
We appreciated the reviewer’s suggestion. We have changed the abstract according to corrections suggested in the manuscript.

Figure 4 is not necessary.

Answer: We strongly agree with the comment. We have removed the figure 4.

Figure 5: It would be interesting to see the actual replication/titer compared to the initial virus that was used for the feeding (mosquito at day 7).

Answer: We performed the vector competence assays as a preliminary investigation and we did the analysis of infection rate (using midguts) and dissemination rate (using salivary glands). We found a dissemination rate of 100% at day 7 and the viral titer ranged from 2.5x103 to 1.25x106 PFU/ml in salivary glands. Therefore, we don't have the exact viral titer of the mosquitoes who transmitted MAYV to mice, but we do have the data from the preliminary study that was used as a reference. We have made an effort to provide the additional information about salivary glands titer.

Please explain line 382-387 (and figure 5)

Did the mosquitoes feed on the mice 3 and 7 dpi with MAYV and then after 7 days the mosquitoes were tested? So figure 5G is a combination of days 3 and 7 feeding? Why merge the days? If there’s a difference/no difference its important with regards to incubation time in mice.

Answer: We have made changes in the text to clarify the order of the experiment and mosquito analysis. On day 3, naive mosquitoes were placed to feed in viremic mice. Seven days later, mosquitoes were collected and analyzed (last part of transmission cycle). We have made changes on figure 5 as requested in the review.

Overuse of regarding and concerning at the start of sentences

Answer: We have revised the text to reduce repeated words and we sent the text to an external English proofing service.

In the discussion, please be more clear when mentioning previous work done. 

Answer: We thank the reviewer for highlighting this requirement. The requested change has been made.

Reviewer 2 Report

Summary

The authors of the paper focus their study on the development of a mice model for the investigation and evaluate the transmission of MAYV by Ae aegypti and Cx quinquefasciatus.

Overall impression/ Broad comments

The research treated in this article point out the development of a mouse model to study arbovirus infection. Authors used MAYV and two species of mosquitoes to evaluate the interest of their mouse model for arbovirus study.

Finding is globally interesting in the development of new model to study arbovirus transmission.

Despite strength points, I’ve some suggestion to improve the manuscript for publication.

Results part.

Table 1 What is the relevance of the titer in only one mosquito? How to statistically evaluate the decay of the titer for a blood meal experiment on 1 mosquito? What is the message of this part of the table? I think MAYV titer is not necessary for the paper if it’s done on only one mosquito.

Figure 3.

The authors present their result in Cq and in RNA copy/ml. The only interesting information is the result in RNA copy number as this result is obtained using the Cq of each specimen for calculation. Fig 3 A and 3 B are not necessary in the manuscript.

In figure 3D, statistical analysis (non-significant) looks curious with a comparison at day 14, of only 3 values widely distributed (with a curious little SEM) for the Culex group compare to more than 40 values for Aedes.

Figure 4 and Line 299-303

This figure is made with the same set of specimens than the fig 3. Why authors did a new graphic instead of using the graph fig 3 to determine. The Fig 4 is not necessary if authors use fig3 the show they have selected day 7 for the time of mice infection.

Figure 5 D: not necessary as authors show the RNA copy number/ml in fig E

Figure 5E: add RNA copy number/ml in the legend of Y axis

Figure 5G Why only the graph of Cq value? Use the graph of RNA copy number/ml instead of Cq, to show the infection of the 2nd batch of mosquitoes on infected mice.

Line 382. Which mice were chosen to be bitten by mosquito for re-infection? There are 12 mice in the test group, which one did authors select to be exposed to mosquitoes?

Minor issues and comments

Please, remove the reference number of every product in Materials and Methods part and homogenate the presentation of reference of product used in this study.

Line 130: modify NB2 by BSL2

Line 140: typo, day not days’

Line 138 and other: homogenate °C in all the manuscript

Line 182: precise the type of diluent, not only a reference

Line 191, Ethic statement at the end of Materials and methods part.

Line 221: add the reference of the caboxymethyl cellulose

Table 1: add the time point of the measurement of the titer of MAYV in the table or in header or in footer legend.

Fig 2: add in figure legend what is graph A) and B)

Line 310 : change the stars for the p value according to the graph. In the graphic there are only 1 or 3 stars. In figure legend there are 1 star or 2 stars two times .

Author Response

Point by point answers:

Table 1 What is the relevance of the titer in only one mosquito? How to statistically evaluate the decay of the titer for a blood meal experiment on 1 mosquito? What is the message of this part of the table? I think MAYV titer is not necessary for the paper if it’s done on only one mosquito.

Answer: We thank the reviewer for this suggestion. During the experiments we analyzed samples as a double check. However, we strongly agree that just one mosquito has no statistical value.

Figure 3. The authors present their result in Cq and in RNA copy/ml. The only interesting information is the result in RNA copy number as this result is obtained using the Cq of each specimen for calculation. Fig 3 A and 3 B are not necessary in the manuscript.

Answer: We strongly agree with this comment. We have modified the figures and removed the graphs of Cq values.

In figure 3D, statistical analysis (non-significant) looks curious with a comparison at day 14, of only 3 values widely distributed (with a curious little SEM) for the Culex group compare to more than 40 values for Aedes.

Answer: We used the Shapiro-Wilk Normality test and our samples did not show normal distribution. Then, the Wilcoxon test was used (non-parametric). This test perform the comparison between the medians of the samples and not the mean. After applying the test, the following p-values were observed in the comparisons between the salivary glands of the two species of mosquitoes in RNA copy number (Figure 3): p=0.1186 (3dpi), p=0.0001233 (7dpi) and p=0.0792 (14dpi). The reviewer's observation is pertinent but our p-values were strictly considered significant when p<0.05. At the 14 dpi mentioned, the p value can be considered as borderline but still not significant.

Figure 4 and Line 299-303

This figure is made with the same set of specimens than the fig 3. Why authors did a new graphic instead of using the graph fig 3 to determine. The Fig 4 is not necessary if authors use fig3 the show they have selected day 7 for the time of mice infection.

Answer: We removed figure 3.

Figure 5 D: not necessary as authors show the RNA copy number/ml in fig E

Answer: We removed the Cq value graph.

Figure 5E: add RNA copy number/ml in the legend of Y axis

Answer: We thank the reviewer for highlighting this and we have modified the figure.

Figure 5G Why only the graph of Cq value? Use the graph of RNA copy number/ml instead of Cq, to show the infection of the 2nd batch of mosquitoes on infected mice.

Answer:  We have modified the graph to RNA copy number/ml.

Line 382. Which mice were chosen to be bitten by mosquito for re-infection? There are 12 mice in the test group, which one did authors select to be exposed to mosquitoes?

Answer: We thank the reviewer for highlighting that this information is unclear. We used groups of four mice as described in table 2 and used all mice with 10-15 mosquitoes per rodent. We described this in the methodology.

Please, remove the reference number of every product in Materials and Methods part and homogenate the presentation of reference of product used in this study.

Answer: We have formatted the details of each item listed in the Materials and Methods part.

 Line 130: modify NB2 by BSL2 - Line 140: typo, day not days’

Answer: Thank you. We have changed it.

Line 138 and other: homogenate °C in all the manuscript 

Answer: We thank the reviewer for the observation. We have changed it.

Line 182: precise the type of diluent, not only a reference

Answer: We have added the components of diluent.

Line 191, Ethic statement at the end of Materials and methods part.

Answer: We strongly agree with the comment. We added the Ethic statement at the end of materials and methods.

Line 221: add the reference of the caboxymethyl cellulose

Answer: We added the reference.

Table 1: add the time point of the measurement of the titer of MAYV in the table or in header or in footer legend.

Answer: We have deleted table 1, as suggested by another reviewer. This information is found in the first paragraph of results topic in “The titer of the initial mixture (MAYV culture + blood) before artificial blood feeding ranged from 2.5x106 to 1.5x107 PFU/mL”.

 Fig 2: add in figure legend what is graph A) and B)

Answer: We thank the reviewer for the suggestion, we did the addition.

Line 310: change the stars for the p value according to the graph. In the graphic there are only 1 or 3 stars. In figure legend there are 1 star or 2 stars two times.

Answer: We thank the reviewer for the observation. The text referred was part of the footnote of figure 4. This figure was removed from the manuscript as suggested by another reviewer.

Round 2

Reviewer 2 Report

The authors of the paper focus their study on the development of a mice model for the investigation and evaluate the transmission of MAYV by Ae aegypti and Cx quinquefasciatus. The research treated in this article point out the development of a mouse model to study arbovirus infection. Authors used MAYV and two species of mosquitoes to evaluate the interest of their mouse model for arbovirus study.

The authors responded to comments on the first draft of the manuscript. The corrections made by the authors improved the quality and understanding of the study.